# Interventions Targeting Bottle and Formula Feeding in the Prevention and Treatment of Early Childhood Caries, Overweight and Obesity: An Integrative Review

**DOI:** 10.3390/ijerph182312304

**Published:** 2021-11-23

**Authors:** Heilok Cheng, Rebecca Chen, Maxim Milosevic, Chris Rossiter, Amit Arora, Elizabeth Denney-Wilson

**Affiliations:** 1Susan Wakil School of Nursing and Midwifery, Faculty of Medicine and Health, The University of Sydney, Camperdown, Sydney, NSW 2006, Australia; jessica.cheng@sydney.edu.au (H.C.); christine.rossiter@sydney.edu.au (C.R.); 2The Westmead Applied Research Centre, Faculty of Medicine and Health, The University of Sydney, Westmead, Sydney, NSW 2145, Australia; rebecca.chen@sydney.edu.au; 3Sydney Dental Hospital and Oral Health Services, Sydney Local Health District, NSW Health, Surry Hills, NSW 2010, Australia; Maxim.Milosevic@health.nsw.gov.au (M.M.); a.arora@westernsydney.edu.au (A.A.); 4School of Health Sciences, Western Sydney University, Penrith, NSW 2751, Australia; 5Health Equity Laboratory, Campbelltown, Sydney, NSW 2560, Australia; 6Translational Health Research Institute, Western Sydney University, Penrith, NSW 2751, Australia; 7Clinical School Child and Adolescent Health, The Children’s Hospital at Westmead Clinical School, Faculty of Medicine and Health, The University of Sydney, Westmead, Sydney, NSW 2145, Australia; 8Sydney Institute for Women, Children and Their Families, Sydney Local Health District, NSW Health, Camperdown, Sydney, NSW 2050, Australia

**Keywords:** dental caries, overweight, obesity, bottle feeding, infant formula, infant health, diet

## Abstract

Overweight, obesity and early childhood caries (ECC) are preventable conditions affecting infants and young children, with increased prevalence in those formula-fed. Previous research has focused on distinct outcomes for oral health and healthy weight gain. However, the aetiology may be linked through overlapping obesogenic and cariogenic feeding behaviours, such as increased sugar exposure through bottle propping and overfeeding. Best-practice bottle feeding and transition to cup use may concurrently reduce overweight, obesity and ECC. This integrative review aimed to identify interventions supporting best-practice formula feeding or bottle cessation and examine the intervention effects on feeding, oral health and weight outcomes. The reviewers searched nine databases and found 27 studies that met the predetermined inclusion criteria. Eighteen studies focused on populations vulnerable to ECC or unhealthy weight gain. All studies focused on carer education; however, only 10 studies utilised behaviour change techniques or theories addressing antecedents to obesogenic or cariogenic behaviours. The outcomes varied: 16 studies reported mixed outcomes, and eight reported worsened post-intervention outcomes. While some studies reported improvements, these were not maintained long-term. Many study designs were at risk of bias. Effective intervention strategies for preventing ECC and child obesity require the holistic use of interdisciplinary approaches, consumer co-design and the use of behavioural change theory.

## 1. Introduction

Overweight, obesity and early childhood caries (ECC) are preventable conditions affecting infants and young children. ECC are dental caries occurring in children aged under 6 years, defined as the presence of one or more primary teeth affected by decay, tooth loss or tooth fillings [1]. Globally, the age-standardised prevalence of untreated caries in primary (baby) teeth of children up to 14 years of age ranged from 4.9% (Australia) to 10.8% (the Philippines) in 2010 [2], with a worldwide prevalence of 7.8% and 126 million age-standardised cases in 2015 [3]. The health consequences of ECC include poor child growth from eating problems and poor nutrition, impaired speech development, and impaired sleep, play, learning, concentration, school performance and attendance due to caries-related pain [4,5,6]. Children who experience ECC are likely to be at increased risk of later dental problems [7]. 

Overweight and obesity in children under 5 years of age, defined respectively as two and three standard deviations above the World Health Organization weight-for-height growth standard median [8], affected an estimated 38.2 million children in 2019 [8]. Longitudinal data show that the trajectory of infant weight gain increases the risk of obesity in childhood and adulthood [9,10]. Obesity during childhood increases the risk of chronic disease, such as type 2 diabetes, cardiovascular disease and non-alcoholic fatty liver disease [11]; children with obesity are over five times more likely to have obesity into adulthood [12]. 

These health conditions may be linked due to overlapping obesogenic and cariogenic feeding behaviours [13]. Increased exposure to sugar increases the risk for both dental caries and excessive caloric intake through various practices: propped bottle feeding in bed for younger infants or bedtime bottle use for older children who can hold bottles [6,14,15,16]; the use of sugar-sweetened beverages in bottles [14]; the addition of fermentable carbohydrates, such as sugar, syrup, honey or cereal, in bottles [16,17]; and frequent exposure to sugar, such as through snacking or drink sipping throughout the day [14,15]. Bottle use past the age of 12 months can entrench constant drink sipping throughout the day, particularly of sugar-sweetened beverages, which can also contribute to dental caries and obesity [15]. Research has found prolonged bottle use at 24 months of age associated with obesity at 5.5 years of age [18]; late bottle cessation after 18.8 months of age is associated with an increased risk of overweight and obesity at 3–5 years old [19]. Two meta-analyses of children aged up to 6 years found an increased risk of ECC for children who were above a healthy weight, although the results were inconsistent across the weight categories of overweightness, obesity and combined overweightness and obesity [13,20]. 

Preventative strategies for both ECC and excessive weight gain include breastfeeding until 6 months of age, avoiding added and free sugar, using responsive bottle feeding, avoiding infant overfeeding, using cups from 6 months of age and eliminating infant bottle use at one year of age [15,21]. A systematic review by Appleton and colleagues identified additional formula feeding practices to reduce the risk of infant overweight and obesity, including choosing infant formula with lower protein content, avoiding ‘follow on’ formulas marketed at infants aged 6 months and above, avoiding the addition of fermentable carbohydrates in bottles and using smaller infant bottles to avoid overfeeding [16].

Healthcare professionals working with infants and their families are well-placed to discuss infant feeding that promotes healthy practices, reduces the risk of overweight and obesity, and reduces cariogenic behaviours [1]. A review on primary preventative oral healthcare for young children or childbearing women and delivered by nurses or midwives found that 14 out of 21 trials reported improved the outcomes of dental caries prevalence, oral health and dietary behaviours and dental service use [22]. Similarly, a review of non-dental health professionals providing preventative dental care found that more effective caries prevention and improved health and dietary behaviours were associated with longer intervention periods with education reinforcement, multiple avenues of verbal and written education and counselling, and comprehensive interventions with education, oral health toolkits and counselling [5]. An oral health model implemented in two Women, Infants and Children (WIC) program centres targeting low-income mothers with children aged under five in the USA demonstrated that allied health clinicians can expand their practices to oral health risk screening, assessment, and fluoride varnish application [23].

Interdisciplinary approaches by healthcare professionals in dental, medical, nursing and allied health settings that address formula feeding and infant bottle use may help to prevent both ECC and the risk of overweight and obesity. Previous research on this topic has existed in two distinct academic silos: that is, dental interventions report outcomes, such as bottle use, as an oral health behaviour but not an obesity risk behaviour and obesity interventions report outcomes, such as the type of fluid consumed, as a dietary but not a oral health risk behaviour. To our knowledge, this is the first integrative review conducted by an interdisciplinary research team to address the dental and nutritional approaches to infant bottle and formula feeding on dental and obesity outcomes. This integrative review aimed to identify interventions, trials and programs undertaken to support best-practice formula feeding or bottle cessation in infants and children and to examine their effectiveness in formula feeding practice, bottle cessation, oral health and/or child weight outcomes.

## 2. Materials and Methods

### 2.1. Research Questions

To capture the breadth of the research across disciplines, this review addressed the questions:What interventions, trials or programs have been undertaken to support best-practice formula feeding or bottle cessation in infants and children, focusing on either oral health or weight-related outcomes?What are the impacts of these interventions on formula feeding practice, bottle cessation, oral health and/or child weight outcomes?

The Preferred Reporting Items for Systematic Reviews and Meta-Analyses (PRISMA) checklist was used to structure the presentation of this manuscript (Appendix A).

### 2.2. Eligibility Criteria

The Population/Problem, Interest and Context framework used to structure the inclusion and exclusion criteria of the search strategy [24,25] is attached as Appendix A. This review included studies focused on interventions aimed at parents or carers of infants and young children to improve formula, bottle or caries-preventing practices that encourage bottle cessation and cup transition and that measured health or behaviour out-comes relating to feeding practices, parent or carer knowledge, and infant anthropometry. 

### 2.3. Information Sources

The database searches were conducted in CINAHL (via EBSCO), MEDLINE (via Ovid), EMBASE (via Ovid), Global Health (via Ovid), Maternity and Infant Care Database, Scopus, ProQuest, PubMed and Web of Science. The searches were conducted in July to August 2020, with weekly checking for relevant updates until June 2021. Hand searching the reference lists and forward citations of the included studies was undertaken to further identify relevant studies [26].

### 2.4. Search Strategy

The Population/Problem, Interest and Context framework was used to devise the search strategy. Eligible studies addressed: (1) interventions on education about infant formula or bottle use, (2) infant formula or bottle use, including bottle or formula cessation, and (3) interventions about infant formula or bottle use and dental caries. An example of the search strategy structured for the CINAHL database, using the Population/Problem, Interest and Context framework, is provided in Appendix A. The search strategies for all the databases are available in Appendix A.

### 2.5. Study Selection

All references were downloaded to Endnote X9. The references were screened by abstract and title for relevance, then assessed for full-text eligibility by author HC. For studies where eligibility was unclear—such as intervention content—the authors discussed their eligibility by using their clinical knowledge and expertise on infant feeding and oral health.

The study selection process is illustrated in a PRISMA diagram (Figure 1). 

### 2.6. Data Extraction and Synthesis

Data on the first author; publication year; country; study aim, design and period or duration; study setting; participants; intervention and comparator group conditions; and study findings were extracted and tabulated. The data was extracted by author HC. The statistical data, where available, were presented as the mean ± standard deviation or median (interquartile range).

### 2.7. Quality Assessment

A quality appraisal of the included studies was undertaken by authors HC and CR using the Mixed Methods Appraisal Tool (MMAT) [27]. The MMAT is an appropriate evaluation tool that accounts for the diverse study types included and rates each article on the specific criteria relevant for its study type. 

## 3. Results

Figure 1 shows the PRISMA study flow diagram. A total of 12,147 references were identified through database searching, and 53 references were identified through hand searching. After removal of the duplicates, 8377 references were screened by the abstract and title for relevance, with 8137 references excluded. Two hundred and forty references were assessed for full-text eligibility. A total of 209 references did not meet the inclusion criteria; of these, 32 references were interventions that were terminated or had minimal formula feeding or bottle cessation support in the wider context of the study intervention. Finally, 31 references were included for analysis, reporting 27 studies, programs, trials or interventions. 

The study findings with outcomes on bottle and cup use, caries prevalence and caries-related dietary behaviours are presented in Table 1.

The summary details of the 32 references that were excluded and which addressed formula feeding, bottle cessation or cup use briefly in the wider context of the study intervention or experienced early study termination are available in Appendix A.

### 3.1. Included Studies

Twelve studies were undertaken in the USA, 4 studies in the UK and 4 studies in Canada. The remaining 11 studies were undertaken in Australia, Germany, Israel, the Netherlands, Syria and Thailand. Thirteen studies utilised a randomised controlled trial (RCT) or cluster-RCT design, five were community-wide programs with quasi-experimental designs and the remaining nine studies were quasi-experimental pre-post-trials or non-randomised controlled trials. Four studies were pilot studies that eventuated into two clinical trials or established evidence for a larger-scale government-run program.

### 3.2. Participants

Eighteen studies were targeted at populations with a specific vulnerability to increased overweight or obesity through infant formula use, dental caries from prolonged infant bottle use, or both: five studies targeted parents or carers in the USA WIC program [28,29,30,31,32,33]; four studies targeted American, Australian and Canadian First Nations communities [34,35,36,37,38,39,40]; four studies were undertaken in communities with social deprivation [41,42,43,44] and five studies were undertaken with cultural groups that report high prevalence of ECC [45,46,47,48,49]. The remaining nine studies targeted parents or carers of infants fed with formula [50,51,52], general populations attending well-child checks [53,54,55] or communities where the prevalence of ECC was reported to be increasing [56,57,58].

Infants and young children as the focus populations of interventions ranged from prenatal parental education [34,38,39,40] to 5 years of age [47]. Most interventions targeted parents as the primary carer [28,29,30,31,32,33,34,35,36,37,38,39,40,41,42,43,44,45,46,47,48,49,50,51,53,54,55,56,57,58]. Five interventions included extended caretakers, such as grandparents, siblings and babysitters [34,35,36,37,47,49].

### 3.3. Interventions

#### 3.3.1. Setting and Interventionists

The studies were undertaken in primary healthcare or community settings or a combination of both. Five WIC-based interventions were implemented during well-child visits or nutrition education classes [28,29,30,31,32,33]. Fourteen studies were undertaken during well-child visits, health visits or vaccination clinics, with physicians, nurses, health visitor (a registered nurse or midwife qualified in public health nursing in the UK [59]), dental healthcare providers and community health workers as interventionists [34,41,42,43,45,46,48,49,51,52,53,54,55,56,57,58]. The *Bottle it up—take a cup!* community program in the Netherlands further targeted staff in day-care centres and playgroups, educators for healthcare workers, and community youth workers [56]. The community-based studies involved community health promotion programs [34,35,36,37,49,56] and/or campaigns [46], or the use of community spaces for education [47] or counselling [38,39,40]. 

The duration of interventions in the primary healthcare setting ranged from one-off education sessions [28,30,31,32,45] or a resource handout [44,45,51] to long-term care embedded in routine well-child visits, health visits or vaccination clinics, with the longest lasting until children were aged 3 years in two studies [41,42]. The community program intervention durations ranged from one year [49] to ongoing, with the *Healthy Smile, Happy Child* project initiated in 1999 and continuing at time of writing [35,60].

#### 3.3.2. Intervention Content: Education

All interventions focused on educating carers during primary healthcare or as health promotion in community programs. The topics encompassed dental caries and oral health [28,30,31,38,39,40,41,48,49,54,55,56,58]; safe infant formula feeding practices [50,51]; infant bottle cessation, transitioning from bottle use to cup use, limiting bottle use and avoiding bottle use at bedtime [29,32,38,39,40,41,42,43,45,46,48,52,53,54,55,56,57,58]; dietary behaviours, including choosing water as a drink, limiting intake and exposure to cariogenic foods and beverages [30,31,38,39,40,41,42,43,44,49,57,58]; healthy weight or growth [28,50]; infant satiety [32]; responsive infant formula feeding [32,33,50]; iron deficiency anaemia [28,54]; tooth brushing [41,43,45,46,49,55,57] and dental registration and attendance [38,39,40,41,44,57]. Although all studies involved education, only six studies described behaviour change techniques [38,39,40,42,47,48,50,52], and only six studies included behaviour changes or educational theory in the intervention design [30,31,32,47,49,50,61].

The behaviour change counselling described in six studies was motivational interviewing, goal-setting or action planning. In three studies, motivational interviewing comprised two telephone calls in addition to outpatient counselling [52], embedding motivational interviewing into the usual health visit processes [48], and four sessions of counselling from pregnancy to 18 months of age [38,39,40]. Goal-setting occurred as part of motivational interviewing [48] or educational classes [47] or through the use of self-directed worksheets [42]. Three studies used action planning to initiate behaviour change goals or to identify and resolve barriers to change [38,39,40,48,50]. 

#### 3.3.3. Intervention Content: Resource Distribution

Resources were used across 17 interventions to facilitate education and behaviour changes. The resources for participants included information pamphlets or handouts [28,33,34,35,43,44,45,51,54,55,57], text messaging [33], child drinking cups [28,29,34,36,37,43,44,45,46,54] and oral health kits, including toothbrushes and/or toothpaste [34,42,43,44,45,46,55,57]. Resources were provided to or developed by interventionists in four studies for patient education and community health promotion [33,35,36,37,56].

#### 3.3.4. Intervention Design and Stakeholder Engagement

Five studies described the design of their interventions [28,29,47,50,56], involving stakeholder engagement for program development. Four studies used focus groups and interviews with interventionists and carers in resource development [61,62], the planning and acceptability of intervention messages [63,64] and determining the acceptability of intervention delivery [64]. 

There were varying levels of community engagement across the studies, such as community-based interventions or community health workers as interventionists. This included community championships to enable education in early childhood caries prevention [35,36,37,49], media, advertising and campaign promotion in the community [34,36,37,46,56], and building capacity in the existing childhood and family community programs and services to deliver intervention activities [34,35,49]. The *Contra Caries* program utilised community-based education classes with community health educators [47]. 

### 3.4. Study Outcomes

The majority of studies reported mixed (16 studies) [29,30,31,32,34,35,36,39,40,42,43,47,48,49,54,56,58] or no statistically significant outcomes (four studies) [28,50,53,55]. Of these, eight studies reported worsened post-intervention outcomes in the intervention group or both the intervention and comparator groups [32,34,35,40,42,43,56,57]. 

#### 3.4.1. Weight or Anthropometry

Of the six studies that reported on weight or anthropometry outcomes, two studies reported an improvement, with significantly fewer children with a BMI above the 85th percentile in one intervention cohort [34] and a decreased risk of rapid weight gain for infants in an intervention trial arm [33]. The remaining studies reported no significant effect on their anthropometric parameters [28,40,50] or worsened effects, with intervention children in two studies reporting significantly greater weight gains than children in the control groups [32,40]. 

#### 3.4.2. Dental Caries Prevalence

The intervention impact on dental caries tended towards improvements or no significant difference across five of the 11 studies. Six studies reported a decreased prevalence of ECC [34,36,37,39,41,42,43]. However, limitations existed in two studies, where a decreased ECC prevalence was found post-intervention at 2 years of age but not at follow-up at 5 years of age [38], and the discontinuation of program counselling and community-based education activities after funding cessation resulted in nonsignificant changes in ECC prevalence in seven American First Nations communities after four years [36]. The evaluation of one community program that only had one cohort for long-term examination found significant differences in ECC prevalence at 4 years of age but not at 2 and 3 years of age [34], which is reflective of the developmental stages presence of tooth eruption and the natural slow progressing nature of dental caries.

Five studies found no significant differences in dental caries prevalence [31,35,49,53,55]. Notably, one program evaluation across four Canadian First Nation communities found a significantly increased prevalence of ECC and severe ECC—where caries patterns were atypical, progressive, acute or rampant [6]—in rural and remote communities compared to an urban community [35]. 

#### 3.4.3. Carer Knowledge and Awareness

In nine studies that reported on carer knowledge and awareness outcomes generally reported improved understanding of bottle feeding [51], ECC [56], bottle cessation [53], oral health knowledge (including topics on bottle use or cariogenic drinks) [30,31,34,35,47], effect of cariogenic drinks on dental health [47,49], and recall of interventionist education [44]; however, it was not significantly improved in one study [58].

#### 3.4.4. Bottle and Cup Use

The outcomes on bottle and cup use were mixed. In ten studies reporting on bottle cessation, seven studies reported an improvement with increased bottle cessation [42,43,45,46,52,56] or earlier cessation [54], while three studies reported no differences in prevalence of bottle cessation or age of cessation [29,34,53]. Two studies reported an earlier start to cup use by 2 to 3 months compared to their comparative groups [34,54]. In nine studies that reported on bottle use or bottle feeding, five studies found a decreased use of bottles in prevalence and frequency of use [29,46,48,54,56], while four studies reported no differences [28,33,35,58]. In three studies that reported on the frequency or prevalence of cup use, two studies found no differences [28,34], and one study found increased cup use [35]. Lawrence and colleagues found lower rates of carer ever using bottles for feeding but no difference in the rates of breastfeeding and no difference in the prevalence of continued bottle use at 2–5 years of age [34]. Ventura and colleagues found that, although exclusive breastfeeding decreased, mixed formula feeding and breastfeeding decreased, and exclusive formula feeding increased from birth to six months; there was no interaction between responsive bottle-feeding intervention strategies and time, indicating that the intervention strategies did not inadvertently promote greater levels of bottle feeding [33].

Bottle use during sleep is a risk factor for dental caries. In four studies with outcomes on bedtime or sleep time use of bottles and cups, one study found an improvement, with a decreased use of non-water cups at bedtime but no difference in non-water bottles at bedtime or bottle cessation at bedtime [42], while three studies found no difference in bottle use during sleep [49,55,57].

Increased exposure to potentially cariogenic drinks through increased bottle or cup uses outside of meal sessions is a risk factor for dental caries. Of two studies, which did not report on which types of drinks were contained in bottles, one study found no difference in bottle use during meals and between meals [57], and the other study found increased bottle use by intervention children outside of meals but decreased bottle use to soothe crying [34]. 

#### 3.4.5. Cariogenic Dietary Behaviours 

The increased frequency of exposure to cariogenic foods and beverages increases dental caries risk. Cariogenic dietary behaviours, external to bottle or cup use, include the frequency of snacking, types of beverages consumed, and the addition of cariogenic foods into infant milk bottles. Across 16 studies, the outcomes on cariogenic behaviours tended towards no significant effects. Twelve studies found no significant differences in the amount or frequency of intake of infant formula [32], milk [29,54], juice [29,31,49,54] and cariogenic or discretionary foods or beverages [29,31,34,40,47,48,49,55,58]; the total fluid intake from bottles and cups [28]; the addition of cariogenic foods to bottles [48]; and the use of sugar- or honey-dipped baby pacifiers [34]. Five studies reported positive outcomes with decreased cariogenic beverage intakes in cups or at daytime [43,58], more children limiting their intake of sweet beverages [47] or foods [41], a decreased addition of sugar or sweeteners to infant foods or bottles [34,58], and an increased use of non-sweetened beverages (cow’s milk and formula) with a decreased use of sugar-sweetened beverages (condensed milk) in one cohort [34]. Two studies found worsened outcomes, with increased snacking between meals [42] or the increased use of bottles with added sugar for feeding [57]. 

#### 3.4.6. Obesogenic Dietary Behaviours 

Bottle feeding can be associated with pressuring feeding behaviours, such as encouraging infants to feed until the bottle is empty, which increases the risk of overfeeding, instead of feeding in response to infant satiety cues. Ventura and colleagues found that responsive feeding styles, pressuring feeding styles or the encouragement of bottle emptying were not impacted by strategic changes to promote responsive bottle feeding in six WIC clinics [33].

#### 3.4.7. Healthcare Professional Practice

Three studies reported changes in healthcare professional practices. Hamilton and colleagues found that more mothers recalled health visitor talking about bottle transition to cup use and limiting sugary food and beverages at the 8-month well-child visit by 39% and 29%, respectively [44]. Strippel and colleagues’ structured oral health education intervention significantly increased the discussion of oral health prevention topics addressed by clinicians during routine paediatric examinations; however, this constituted half of the 15 topics outlined in the intervention protocols being addressed [58]. The evaluation of the *Bottle it up!—take a cup* campaign found that more child health staff discussed transitioning from bottle to cup use (from 15% to 27%), but fewer child health staff warned against incorrect bottle use (75% to 24%) [56].

### 3.5. Critical Appraisal

A critical appraisal with the MMAT is summarised in Appendix A. 

Of 13 studies (16 references) with an RCT or cluster-RCT design, almost all (11 out of 13) had unclear risks of bias on adherence to the assigned intervention, with adherence not reported. Six studies were at a high risk of bias from the inadequate description of participant randomisation. Five studies (six references) were at a high risk of bias with the groups not being comparable at the baseline: differences in the risk of overweight or obesity [28], cultural background [43], socioeconomic status [45], maternal age [45], infant feeding intentions [54] and registration for private health insurance [30,31]. Only six studies retained ≥80% of participants at the intervention end or follow-up and were at a low risk of bias for incomplete outcome data. Most studies had outcome assessors blinded to the participant conditions, with only four studies at a high risk of bias from unblinded assessors. 

Of 14 studies (15 references) with a quasi-experimental non-randomised study design, almost all (13 out of 14) used appropriate measurements for measuring intervention exposure and participant outcomes. Eleven studies (12 references) involved participants who were representative of the target population, while the remaining three studies provided minimal information about parent demographics. Six studies were at a high risk of bias for incomplete outcome data, with ≤80% retention of participants at the intervention end or follow-up. Nine studies (10 references) were at a high risk of bias, as confounders were not accounted for in the designs and statistical analyses. There was a mixed risk of bias of adherence to the intervention administered.

## 4. Discussion

This integrative review synthesised 27 studies that investigated formula feeding or bottle cessation in infants and young children, and their effects on anthropometry, caries prevalence and dietary behaviours. This review assessed the effectiveness to date of interventions with a dual focus on oral health and child weight outcomes. A range of intervention strategies were used, primarily focused on education and resource distribution. Education for carers addressed various topics on oral health, feeding and dietary behaviours, dental care attendance and tooth brushing. The resources used to facilitate health behaviours included information handouts for carers and interventionists, children’s drinking cups and oral health kits with toothbrushes and toothpaste. Intervention effectiveness was mixed: most studies reported mixed or non-statistically significant outcomes, and eight studies demonstrated worsened post-intervention outcomes. Notably, over half of all studies were targeted at infants and young children with risk factors predisposing them to ECC and excessive weight, including financial or social disadvantages, cultural factors and inequity related to First Nations backgrounds.

This literature review has several strengths. First, it integrates research from multidisciplinary research streams that are usually separate but that have parallel goals. The findings and recommendations are relevant to strengthen practices and develop effective interventions across disciplines. Clinical practices used frequently in certain disciplines—for example, the distribution of child drinking cups to support bottle cessation—can improve intervention designs. Second, it demonstrates the importance of interdisciplinary practice, particularly where interventions are focused on vulnerable populations at increased risk of dental caries, overweight and obesity in early childhood. Third, this review adopted a comprehensive search strategy identifying over 12,000 references, although it is possible that some may have been missed, especially older publications and smaller-scale interventions in local areas not included in peer-reviewed publications.

The limitations of this review include the inability to draw definitive conclusions or conduct meta-analyses due to the breadth or the diversity of study designs and reported outcomes. The study quality varied, with 21 studies displaying a high risk of bias in at least one study dimension, most frequently due to incomplete outcome data (seven RCT/cluster-RCT studies and six non-randomised experimental trials) and in confounders not being accounted for in the design and analysis (nine experimental trials). Furthermore, the impact of these interventions was mixed. Moreover, evaluation with the MMAT indicates limitations in study designs, which may contribute to the misestimation of differences between the intervention and comparator outcomes. Only two studies reported improvements in anthropometry, with fewer children who were overweight or at risk of being overweight in one of two participant cohorts [34] and a decreased risk of rapid weight gain for infants in an intervention arm [33]. Of 11 studies with caries outcomes, six reported improvements; however, of these studies, two experienced no long-term effects after funding cessation or trial completion [36,38], and four reported a high risk of bias for incomplete outcome data [34,39,41,43]. Carer knowledge and awareness of dental caries and feeding behaviours were reported as the most consistent improvements in nine studies. Across 16 studies on cup and bottle use, 11 studies found increased cup use, reduced bottle use and earlier bottle cessation or start of cup use; however, nine of these studies reported a high risk of bias, most commonly in comparable baseline data, complete outcome data and confounders in their designs [29,33,34,43,45,46,52,54,56]. Few clinically significant changes were found for bottle use during sleep or bottle use outside of meal sessions as a potential contributor for cariogenic exposures. Across 16 studies with outcomes on cariogenic dietary behaviours, only five studies had positive outcomes with a decreased exposure to cariogenic foods and beverages but also reported a high risk of bias across at least one MMAT domain [34,41,43,47,58]. Whilst the breadth of the outcome measures did not enable a meta-analysis, the range of the study findings indicates strengths present in both disciplines that should be used to inform future research and intervention designs. Utilising or establishing core outcomes in infant feeding, dietary intake or oral health interventions [65,66] can support the standardised reporting and comparison of meaningful effects.

Health-related behaviour changes are difficult. Behaviour changes should not be dependent on individual-level actions—such as expecting information and knowledge to change established behaviours—without also understanding the underlying factors [67]. Interventions informed by behaviour change theory and that address multifaceted factors underpinning behaviours may support long-lasting changes [68]. Although many studies focused on parent or carer education and/or resource distribution [28,29,30,31,32,33,34,41,43,44,45,46,51,53,54,55,57,58], only six studies utilised motivational interviewing, action planning or goal setting in addition to education [38,39,40,42,47,48,50,52]. Similarly, only six studies included behaviour changes or educational theory [30,31,32,47,49,50,61]. Antecedents of obesogenic and cariogenic bottle, beverage and formula-feeding behaviours relate to knowledge gaps and cultural preferences, including child soothing or settling [69,70], increasing weight gain from perceived poor appetite [71,72], the preference for children with larger body sizes [73,74], and misconceptions on the cariogenicity of drinks [71,75,76]. Barriers to non-cariogenic drink consumption, such as water, by infants and young children may include a child’s dislike of water, a child’s preference for sweet cariogenic drinks, a carer’s concerns about the safety of tap water, a carer’s belief that drinking water shows poverty and the inability to purchase drinks, a carer’s belief that milk is a meal instead of a drink, and norms that do not support drinking tap water [75,76,77]. Without addressing these factors, education focusing on how to undertake best-practice behaviours may be insufficient to promote improved oral health and feeding behaviours. 

To support behaviour changes, motivational interviewing as a client-centred approach can address underlying behaviours and develop parent-directed goal setting. Three studies in this review utilised motivational interviewing as part of carer education [38,39,40,48,52] and found some significant improvements in bottle use and ECC prevalence. A meta-analysis by Borrelli and colleagues found that motivational interviewing targeting parents or parent–child dyads in health interventions improved children’s oral health hygiene, physical activity, screen time use and diet [78]. In an oral health context, motivational interviewing is typically delivered in clinical settings; however, this is often too late—as children present for clinical care after experiencing caries-associated pain—and is not feasible as a population-wide strategy, as it is time- and labour-intensive. Further, although prevention behaviour change interventions have included patient-focused dietary and oral hygiene counselling delivered alongside operative clinical interventions in clinical settings, these interventions are time- and workforce-intensive, expensive to deliver and, without regular and repeated exposure, have shown inconsistent results on sustainably improved dental caries outcomes [79,80]. This provides the impetus for collaborative and interdisciplinary approaches for disease prevention prior to children presenting for clinical treatment. 

The maintenance of these interventions and long-term follow-up are essential. A longitudinal study design in future interventions should be considered, as dental caries is a progressive disease. This was exemplified in two studies: one cohort of an First Nations Canadian community program that found significant difference in ECC prevalence at 4 years of age but not at 2 and 3 years of age [34] and a RCT with First Nations Australian children that found a significant decrease in ECC prevalence at 2 years but not at 5 years of age [38,39,40]. Furthermore, this integrative review highlights the importance of behaviour change theory underpinning long-term intervention designs, as well as the importance of interdisciplinary approaches alongside consumer involvement when developing holistic long-term health education interventions. 

Consumer involvement is essential in developing appropriate messages and strategies acceptable to target communities. Five studies [28,29,47,50,56] involved user engagement as part of the intervention design of resources [61,62], intervention messages [63,64] and intervention delivery [64]. Equally essential is community ownership and participation. First Nations community members and health workers in Australia, Canada and the USA were engaged in building the organisational capacity, delivering interventions, acting as community champions, and implementing local solutions in four oral health promotion programs [34,35,36,37,38,39,40]. Likewise, in-group community health workers across three studies supported program delivery to culturally and linguistically diverse communities in Canada and the USA [46,47,48]. Community engagement does not guarantee successful outcomes, with one study concluding that a culturally sensitive health promotion intervention did not warrant service-based implementation [40]; however, this remains important to the study design and may contribute to how *Healthy Smile, Happy Child* remains an ongoing community-led program in Manitoba, Canada [81]. Future interventions undertaken in populations vulnerable to ECC and child overweight and obesity should involve user co-design of interventions, particularly where cultural appropriateness is the key in designing intervention messages and supporting behaviour changes [82].

Interdisciplinary approaches to the prevention of ECC and child overweight and obesity by addressing best-practice formula feeding and bottle use can strengthen and focus preventative care. In the authors’ local health area, an Early Childhood Oral Health Program integrates oral healthcare into general health interventions by child health professionals [83,84], and the surgical treatment of ECC in public hospitals requires attendance with an oral health therapist and dietitian in ECC prevention education. A midwifery-initiated oral health service with antenatal dental treatment in the Greater Western Sydney region of Australia improved maternal oral health knowledge, oral hygiene and health and the uptake of dental services, where the process evaluation reported positive experiences by midwives, dental health professionals and mothers [85,86,87]; further, it has since developed state government prenatal oral health resources, been adopted into a policy in the state of Victoria, and been integrated into the national body of midwifery’s continuing education program [88,89]. Similarly, the *Healthy Tums Healthy Gums* program, delivered by social workers, oral health staff and dietitians to vulnerable families, improved oral health and childhood nutrition knowledge, including cup use from 6 months onwards, the cessation of bottle use by 12 months of age, and identifying cariogenic and non-cariogenic foods and drinks [90]. An integrated obesity and ECC prevention approach to promoting best-practice formula feeding and bottle use, as exhibited in emerging research [48,91,92], is a promising novel approach that addresses the risk factors identified by child health professionals as contributing to obesogenic formula-feeding behaviours [93]. 

## 5. Conclusions

This integrative review of 27 studies combined research from disciplines that share similar goals regarding infant formula and bottle use, with implications for long-term metabolic and oral health outcomes. The intervention strategies in primary healthcare, community settings and a combination of both ranged from one-off education sessions or resource handouts to long-term care embedded into the usual care practices. While intervention effectiveness was mixed and most studies reported mixed or non-statistically significant outcomes, a range of intervention strategies was demonstrated. This included education, behaviour change counselling, resource distribution and stakeholder engagement.

The findings and recommendations of this integrative review are relevant to strengthen practices by emphasising the need for collaborative interdisciplinary approaches that incorporate dental and nutrition messages to prevent ECC and child overweight and obesity. Specific disciplinary strategies, such as targeted resource use for supporting behaviour changes, should be used to develop effective interventions across disciplines. This review emphasised the need to use behavioural change theory, stakeholder involvement and co-design in intervention development in order to support vulnerable populations at combined increased risks of dental caries, overweight and obesity in early childhood. 

## Figures and Tables

**Figure 1 ijerph-18-12304-f001:**
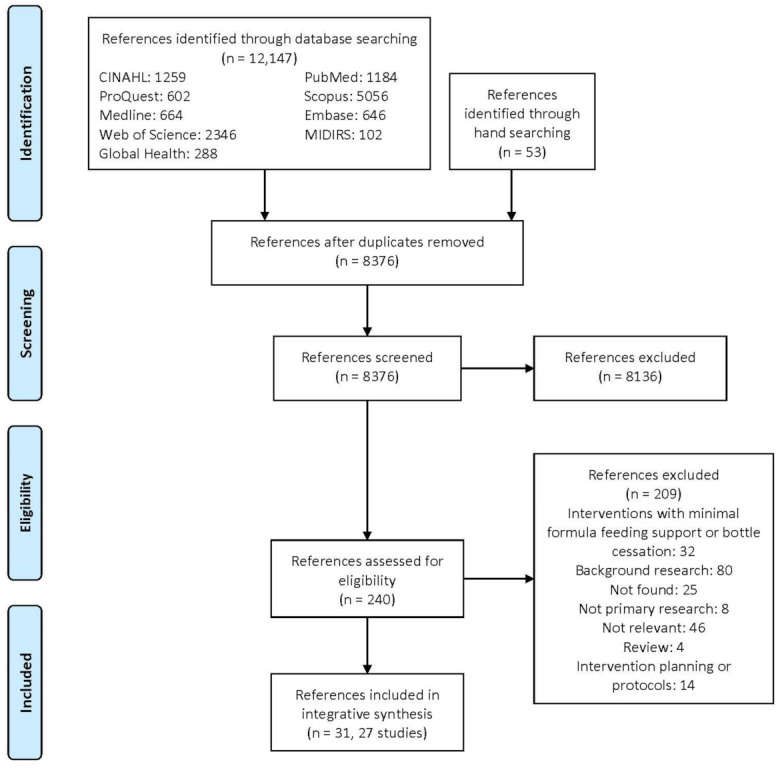
PRISMA flow diagram of the included and excluded references.

**Table 1 ijerph-18-12304-t001:** Summary of the included studies (*n* = 27 studies, 31 references).

Reference. Country	Aim	Study Type; Period	IG and CG Sample	IG and CG Description	Key Findings
Bonuck et al. 2014. USA.	To evaluate the impact of a WIC-based counselling intervention on bottle use and WFL.	RCT; 2008–2011.	Infants aged 11–13 months, consuming > 2 bottles of milk or juice daily, in USA WIC program. IG: *n* = 149 at baseline; 49 after 12 months. CG: *n* = 150 at baseline; 55 after 12 months.	IG: Resources distributed: pamphlet, sippy cup. Education, during 12 months. WIC visit: healthy weight, ECC, iron deficiency, based on precaution adoption process model. CG: Regular WIC care.	At 24 months: NS difference in risk of overweight (WFL > 85th percentile). NS difference in mean daily number of bottles used; mean daily number of sippy cups used; prevalence of any bottle use, prevalence of any sippy cup use; mean fluid intake from bottles; mean fluid intake from sippy cups.
Borghese-Lang et al. 2003. USA.	To develop an informational handout on FF and assess parental evaluation of its usefulness.	Quasi-experimental trial; April–May 2002.	Exclusively FF infants from birth to 4 months. IG: 31 handouts distributed; 22 parents followed up by telephone. CG: N/A.	IG: Development of information handout on successful formula feeding, with 5.5 grade reading level, and distributed to parents at well-baby visits. CG: N/A	19 (86%) read handout. 18 (of 19) increased understanding of bottle feeding after reading handout.
Boonrusmee et al. 2021. Thailand.	To evaluate a novel telephone-based intervention on bottle cessation.	RCT; January 2018–March 2019.	Bottle-fed children, 21–24 months, attending routine well-child visit. IG: 51; 46 completers. CG: 51; 46 completers.	IG: Telephone call at 3 and 6 weeks after receiving routine outpatient advice: motivational interviewing on bottle cessation, individual counselling to support weaning. CG: Routine outpatient advice on bottle cessation.	At 8 weeks: Increased bottle cessation: 41.3% vs. 17.4%, *p* = 0.022.
Braun et al. 2017. USA.	To assess an oral health promotion intervention for medical providers’ impact on ECC in children aged 36–42 months.	Quasi-experimental trial; 2009–2015.	8 Denver Health federally qualified health centres—89% of paediatric patients living below federal poverty line. IG: *n* = 1646 in 2011, receiving fluoride varnish application; *n* = 1708 in 2015. CG: 2009 pre-intervention cohort. *n* = 1501, receiving fluoride varnish application.	IG: ECC risk assessment and fluoride varnish, from 6 months to 3 years. Resources distributed: oral health kit with toothbrush, toothpaste. Education: bottle cessation, water intake, limit snacking. Self-management goal sheet handout. CG: Pre-intervention usual care.	Post-program evaluation of children from 2009 (pre-intervention), 2011 (mid-intervention) and 2015 cohorts. At 36–42 months: Decreased ECC: 46.5%, 57.6% and 37.3%, *p* < 0.001. Increased bottle cessation: use of bottles in 7.9%, 8.1% and 2%, *p* = 0.03. Decreased use of non-water sippy cups in bed: 87.1%, 76.0% and 78.4%, *p* = 0.005. NS difference in non-water bottle use in bed; bottle cessation at bedtime. Increased snacking between meals: 59.9%, 64.8% and 75.2%, *p* = 0.01.
Bruerd et al. 1989. Bruerd et al. 1996. USA.	To prevent baby bottle tooth decay in 12 First Nations American communities, by 50% over 5 years.	Experimental community program; 1986–1989, with follow-up in 1994.	First Nations American communities. IG: 2 × 4 pilot sites, at high and medium intensity, intensity varied on training of interventionists. *n* = 928 screened in 1985; 932 screened in 1989. CG: 4 pilot sites, at low intensity. *n* = 455 screened in 1985; 640 screened in 1989.	IG: Community championship for parent education. Resources distributed: counselling booklets, two-handled cups, posters, stickers. Education: bottle cessation and cup use by 1 year age. Media and advertising. CG: Site coordinators received educational materials but no training.	After 4 years in 1989: 33%, 18% and 27% reduction in prevalence of ECC in high-, medium- and low-intensity intervention sites (*p* < 0.001 each). Overall, 25% reduction in ECC across all sites (*p* < 0.001). At 4 years follow-up in 1994 (after funding cessation in 1990), compared with 1986 baseline: - 25% reduction in ECC across all sites (*p* < 0.001). - 38% reduction in ECC, in 5 sites with ongoing program (*p* < 0.001). - 13% reduction in ECC, in 7 sites who ceased program in 1989 (NS, *p*-value NR).
Cheng et al. 2019. USA.	To modify oral health behaviours after a nursing intervention targeted at children with ECC risk.	Retrospective longitudinal study, intervention and annual training in 2010–2015, clinical records April 2013–June 2015 evaluated.	Children, 9 months to 4.9 years age, with ≥2 documented ECC risk assessments, attending well-child visits across two urban clinics. *n* = 2097.	IG: Fluoride varnish applied. Education targeted to risk factors, using flipchart: caries aetiology, sugar content, bottle/sippy cup use, teeth brushing instructions with adult supervision. Resources distributed: toothbrushes and toothpaste. CG: N/A, pre-intervention care.	From visit 1 (baseline) to visits 2 and 3 (follow-up): ~33% children with non-water bottle/sippy cup use in bed at visits 1, 2 and 3. NS increase in SSB intake, *p* = 0.05: - 1–2 serves/day, 37.2% vs. 41.6% and 43% - ≥3 serves/day, 11.5% vs. 15.6% and 15.7%. NS difference in frequency of ECC: 3.5% vs. 4.7% and 3.5%, *p* = 0.413.
Davies et al. 2005. UK.	To assess the effects of a multi-stage dental health promotion programme in reducing ECC.	Cluster RCT; 1999–2003.	8 month infants, attending health development checks, in socially deprived inner-city area without fluoridated water. IG: randomly sampled children from a Primary Care Group. *n* = 649 → 79 respondents at 21 months; 190 at 3–4 years assessment. CG: randomly sampled children. *n* = 558 → 89 respondents at 21 months; 148 at 3–4 years assessment.	IG: Resources distributed: toothpaste, toothbrush, education leaflet, trainer cup. Education: bottle cessation, cup use, safe drinks, teeth brushing instructions. Intervention at 8 months, 12–15 months, 18 months, 26 months and 32 months of age. CG: Usual care and standard development checks.	At 21 months: Increased bottle cessation: 33% in IG vs. 18% in CG, *p* = 0.04. Increased limited bottle use only at bedtime, by bottle-using children: 43% in IG vs. 62% in CG, *p* = 0.02. Decreased cup use for only drinking unsafe (cariogenic) drinks: 13% in IG vs. 30% in CG, *p* = 0.02. NR difference in non-cariogenic drink use in bottles; trainer cup use at 21 months; cup use for combined cariogenic and non-cariogenic drinks. At 3 to 4 years: Decreased prevalence of ECC at <3 years age: 16.6% in IG vs. 23.5% in CG, *p* = 0.003. Decreased prevalence of ECC at ≥3 years age: 28.7% in IG vs. 39.2% in CG, *p* = 0.001.
Franco et al. 2008. USA.	To evaluate the effectiveness of intensive counselling on bottle cessation.	RCT; September 1999-June 2002.	4 month infants, predominantly AA, attending well-child visit. IG: *n* = 67 parent/infant dyads at 12 months. CG: *n* = 65 parent/infant dyads at 12 months.	IG: Education at 4 months, 6 months, 9 months and 12 months: bottle cessation, transition to cup use at 9 months. CG: Education: brief counselling on cup use at 6 months, bottle cessation at 9 months and 12 months.	At 12–24 months: NS difference in prevalence of ECC; bottle cessation. Increased knowledge of bottle cessation by 12 months age: 49% of IG parents vs. 68% of CG parents, *p* = 0.049.
Hamilton et al. 1999. UK.	To evaluate an oral health promotion programme provided by HV, directed at mothers of 8 month infants.	Quasi-experimental trial; cross-sectional surveys with historical CG Nov 1996 and IG Nov 1997.	Mothers of ~1 year children, seen by HVs from a community healthcare centre in a deprived inner-city area. IG: *n* = 182 mothers, randomly selected from Child Health Register. CG: *n* = 170 mothers, randomly selected.	IG: Resources distributed at 8 month well-child visit: feeder cup, toothbrush, toothpaste. Education: information handout, re: safe drinks for infants, sugar-free medication, dental registration. CG: N/A, pre-trial usual care.	Increased mothers recall HVs talking about using feeder cup instead of bottle: 93% post-trial vs. 54% pre-trial, *p* < 0.001. Increased mothers recall HVs talking about limiting sugary food and drink: 91% post-trial vs. 62% pre-trial, *p* < 0.001.
Harrison et al. 2003. Canada.	To design, implement and evaluate an oral health promotion program for inner-city Vietnamese preschool children.	Community program; 1994–2001 (approx.).	Vietnamese mothers with infants attending well-child visits. IG: mothers-child dyads followed up after counselling clinic; *n* = 25, 1994; *n* = 25, 1998; *n* = 17, 2001. CG: children of a similar age from neighbouring Vietnamese community. *n* = 14 baseline, *n* = 9 comparison children.	IG: Education, during 2 month, 4 month, 6 month, 12 month and 18 month well-child check: avoid bottle use in bed, teeth cleaning, transition to cup use. Resource distribution: toothbrush, training cup. Community outreach. CG: No intervention.	At 1996 follow-up clinic, at children 18 months age: For IG children, compared to baseline and comparison children: - decreased daytime bottle use: 6.3% vs. 83.3% and 55.6%, *p* < 0.05. - decreased bottle use in bed: 13.3% vs. 69.2% and 55.6%, *p* < 0.05. For IG cohorts at 1994 baseline and 1998 and 2001 follow-up: - decreased daytime bottle use: 81.8%, 0% and 11.8%, *p* < 0.05. - decreased bottle use in bed: 66.7%, 12.0% and 11.8%, *p* < 0.05. - cessation of bottle use by 2 years: 13%, 83.3% and 88.2%, *p* < 0.05.
Hoeft et al. 2016. USA.	To determine program effectiveness of parents’ oral health knowledge and behaviours for their young children.	Quasi-experimental trial; August-December 2011, final survey follow-up March 2012.	Low-income Spanish-speaking parents/carers of children aged 0–5 years. IG: *n* = 105 enrolments, with *n* = 95 post-intervention and *n* = 79 3 months post-intervention. CG: N/A.	IG: Education: caries aetiology, toothbrushing behaviours, reducing sugar intake, snacking and bottle use, dental visits. Behaviour management and goal setting on tooth brushing. CG: N/A.	13 children using a bottle at baseline, with 3 children ceasing bottle use by 3 months post-intervention. Increased children having limited (<1/day) sweet drink intake: 33% baseline vs. 77% post-Intervention, *p* = 0.0082. NS long-term difference: 77% post-intervention vs. 63% 3 months post-intervention, *p* = 0.1306. NS difference in limited (<1/day) sweet food intake, across baseline, post-intervention and 3 months post-intervention. Increased oral health knowledge (scored out of 16): 12.8 ± 1.6 baseline vs. 15.2 ± 0.7 post-intervention, *p* < 0.0001. NS difference in oral health knowledge post-intervention and 3 months post-intervention. Includes: increased correct response that sippy cup for milk consumption at bedtime is bad for children’s teeth: 66% baseline vs. 96% post-intervention.
Joury et al. 2016. Syria.	To investigate the impact of an integrated oral health promotion intervention, within a national immunisation programme, on tooth-brushing and bottle-feeding termination practices.	Pilot RCT: 2 parallel CGs, 1 IG; March-May 2013.	Mothers of 1 year old infants, attending an infant vaccination clinic. IG: *n* = 32. CG 1: *n* = 30. CG 2: *n* = 30.	IG: Resource distribution: trainer cup, toothbrush, toothpaste. Education: information pamphlet on bottle cessation, cup use, tooth brushing. CG 1: Oral health information pamphlet. CG 2: No intervention.	100% infants bottle feeding at baseline. Increased bottle cessation at 1 month follow-up: 18.8% bottle use in IG vs. 69.2% in CG 1 and 93.8% in CG 2, *p* < 0.001.
Kahn et al. 2007. USA. (Pilot study to Bonuck et al. 2014).	To pilot test a standardised protocol for bottle cessation by parents in WIC program.	Pilot RCT; dates NR.	Infants aged 18–30 months, who consumed >3 bottles daily, attending WIC program. IG: *n* = 18 retained at follow-up. CG: *n* = 21 retained at follow-up.	IG: Education: parental style, parent’s feelings about bottle use, bottle cessation protocol, usual WIC care. Resources distributed: sippy cup. CG: Usual WIC care.	Decreased daily bottle use: 0.9 in IG vs. 2.2 in CG, *p* < 0.05. NS difference in bottle cessation; type of beverage consumed in bottles (milk, juice, sweet beverages).
Karasz et al. 2018. (Pilot study to Karasz et al. 2018) USA.	To pilot the acceptability and feasibility of an oral health prevention program, conducted by community health workers, for South Asian children at high risk of ECC.	Pilot RCT; dates NR.	Bangladeshi immigrant mothers with a 6–18 month infant. IG: *n* = 38, 31 at follow-up. CG: *n* = 21, 21 at follow-up.	IG: Enhanced usual care: 2 home visits, 3 follow-up phone calls after 3 months, 6 months, 9 months. Motivational interviewing and education: oral health education, set goals on change on bottle feeding, develop action plan. CG: Enhanced usual care. Education: oral health pamphlet, 5 min oral health counselling, referral to local dentists.	Decreased total and nap/night-time bottle intake in IG vs. CG, *p*-value NR, *p*-value significant. NS decrease in total bottles with added solids/sweeteners; SSBs; sweets. 150% increase in bottle use in CG from baseline; 36% decrease in IG from baseline.
Kavanagh et al. 2008. USA.	To evaluate whether education about infant satiety cues would alter FF practices and infant formula intake and weight gain.	RCT; dates NR.	Carers of exclusively FF infants aged 3–10 weeks, attending WIC. IG: *n* = 44, 19 completed final data collection. CG: *n* = 57, 21 completed final data collection.	IG: Education, replacing usual WIC class: infant feeding, avoid preparation of excessive formula, awareness of infant satiety, discouraging use of >6 oz formula bottles. CG: General guidance on infant feeding.	At 20–27 weeks: Increased adjusted mean weight: 7214 g in IG vs. 6758 g in CG, *p* = 0.006. Increased weight gain per week: 195.3 in IG vs. 156.1 g in CG, *p* = 0.008. Increased adjusted mean length: 64.2 cm in IG vs. 63.3 cm in CG, *p* = 0.02. Increased length gain per week: 0.70 cm in IG vs. 0.63 cm in CG, *p* = 0.045. NS difference in final formula intake (mL/24 h), change in formula intake from baseline to end of study, % of bottles emptied at baseline and end of study and % of >6 oz bottles offered at baseline and end of study.
Koelen et al. 2000. The Netherlands.	To evaluate the effect of an ECC prevention, implemented a nationwide scale, on increasing knowledge and awareness in HCPs; motivate caries prevention discussions from HCPs to parents; increase parental awareness of prolonged bottle use and increase transition from bottle to cup at 9 months.	Quasi-experimental national trial; May 1995–1997.	Parents of 0–4 year children, specifically in 9–18 month range. IG: *n* = 60 local child health centres contacted for post-program evaluation → 40 centres, 73 HCPs and 102 parents (53 participating in pre-program evaluation) participated. CG: *n* = 20 local child health centres contacted for pre-program evaluation → 16 centres, 22 HCPs and 135 parents participated.	IG: Campaign in primary, secondary and tertiary services. Resources distributed for interventionists and parents: colouring and message sheets; visual resources; tear-off pads. Education by dental hygienists and oral health workers. CG: N/A.	At 1.5-year program evaluation: Increased discussion by HCPs on transitioning from bottle to cup: 27% post-program vs. 15% pre-program. No change in discussion of ECC by HCPs: ~75% pre- and post-program. Decreased HCPs that warn against incorrect bottle use: 24% post-program vs. 75% pre-program. Increased parental awareness of ECC: 78% post-program vs. 60% pre-program, *p* < 0.05. Decreased parental use of bottles: 88% vs. 64% of new parents, *p* < 0.001. Decreased frequency of bottle use, post-program compared to pre-program: - 1/day, 34% vs. 6% - 2/ day, 31% vs. 21% - 3/day, 11% vs. 33% - >3/day, 23% vs. 39%, *p* < 0.001. Increased frequency of switching from bottle to cup before 12 months: 88% post-program vs. 72% pre-program, *p* < 0.1 (NB: *p*-value as reported by authors).
Kowash et al. 2000. UK.	To determine the effect of dental health education on caries incidence in infants, through regular home visits by trained dental health educators over a period of 3 years.	Quasi-experimental trial with cohort design of interventions of varying intensity; dates NR.	Mothers with children born between Jan and September 1995, residing in a deprived area with high caries prevalence. IG A: *n* = 60. IG B: *n* = 59. IG C: *n* = 60. IG D: *n* = 40. CG: *n* = 55.	IG: Intervention A or B: dental health education, focused on diet or oral hygiene. Intervention C: Intervention A + B. Intervention A–C: 15 min dental health education at HV every 3 months from 0–2 years, then twice/year for 1 year. Intervention D: Intervention C, but only one year home visit. Education: replace bottle with cup, tooth brushing, regular dental attendance. CG: No intervention.	At 3 years: CG children with more frequent consumption of sweets, *p* < 0.001, c.f. all IG: - 33% in CG, vs. 9% in IG A, 0% in IG B, 2% in IG C and 8% in IG D, with >1/day sweet consumption - 5% in CG, vs. 58% in IG A, 64% in IG B, 45% in IG C and 50% in IG D, with 1/week sweet consumption Decreased ECC prevalence: 2 (4%) in IG A vs. 18 (33%) in CG, *p* < 0.001. IG children with more frequent limited sweet consumption, after meals only, or on weekends only: 33% in CG, vs. 75% in IG A, 70% in IG B, 63% in IG C and 62% in IG D, *p*-value NR.
Lakshman et al. 2018. UK.	To assess the efficacy of a theory-based behavioural intervention to prevent RWG in FF infants.	RCT; March 2011–June 2015.	Healthy, full-term infants FF within 14 weeks of birth. IG: *n* = 340 at baseline, 293 at 12 mo. CG: *n* = 329 at baseline, 293 at 12 mo.	IG: 3 face-to-face and 2 phone contact visits with nurse facilitator until infant 6 months. Education: reduce formula intake; use responsive feeding; monitor growth. Action planning, goal setting and coping planning. CG: Other FF education: equipment sterilisation, formula preparation, parenting, sleep.	At 6 months: Reduced milk intake: 836.1 mL/day in IG vs. 895.9 mL/day in CG. Difference = −59.7 (−91.1 to −28.3), *p*-value NR. At 12 months: NS mean change in weight SDS; BMI SDS; RWG prevalence.
Lawrence et al. 2004. Canada.	To evaluate the effectiveness of the dental hygiene-coordinated prenatal nutrition program, delivered by community-based nutrition educators, on dental health and child feeding habits; child oral health status; and early childhood obesity.	Cross-sectional longitudinal evaluation of community program; launched mid−1996.	Carers of infants born in June 1996-Feburary 1999, in the First Nations reserve. IG: *n* = 230 in 2001, 215 responses; *n* = 367 in 2002, 217 responses. CG: *n* = 241 in 2001, 182 responses; *n* = 338 in 2002, 158 responses.	Resources distributed: two-handled cup, toothbrush, dental information sheets. Education, in prenatal program: nutrition and dental prevention. Community campaign. IG: 4 ‘high intensity’ sites with ≥70% distribution of program’s oral health promotion materials to mothers. CG: 4 ‘low intensity’ sites, with ≤10% distribution of program’s oral health promotion materials to mothers.	At 2–5 years, comparing high-intensity sites to low-intensity sites in 2001 and 2002: Increased oral health knowledge (including topics on bottle use for child soothing; SSB, formula or milk damaging children’s teeth; and ad libitum bottle feeding of older children) in high-intensity group in 2001 (*p* < 0.001) and 2002 (*p* < 0.05), c.f to low-intensity group. Decreased rate of ever bottle feeding, 78.4% and 80.1% vs. 86% and 88.6%, *p* < 0.05; however, NS difference in rates of breastfeeding. NS difference in age of bottle-feeding initiation; number of sugary snacks eaten per day; use of sugar/honey-dipped baby pacifiers; age of bottle cessation; continued bottle use at 2–5 years. Increased bottle use outside of meals by high-intensity group: during nap time, *p* = 0.01; and bedtime, *p* = 0.018, in 2001. Decreased bottle use outside of meals by high-intensity group: ‘always’ during child crying, 15.9% vs. 31.2%, *p* = 0.009, in 2002. NS difference in BMI weight categories in 2001. Increased normal weight children, 54.5% vs. 40.5%; decreased ‘at risk of overweightness’ children (BMI ≥ 85th to <95th), 16.2% vs. 24.7; and decreased overweightness children (≥95th BMI), 25.8% vs. 31.8%, *p* = 0.023, in high-intensity vs. low-intensity group in 2002. For 2001 respondents: Similar prevalence of ECC at 2 years and 3 years age; decreased ECC at 4 years age: 49.1% in high-intensity vs. 67.8% in low-intensity group, *p* < 0.05. For 2002 respondents; Trend towards decreasing unregulated bottle sipping by child during the day in high-intensity vs. low-intensity group, *p* = 0.052. Cariogenic fluid use in bottles similar, with decreased sugar-sweetened cariogenic fluid use in bottles, in high-intensity vs. low-intensity group: cow’s milk, 28.1% vs. 14.3%; condensed milk (sweetened evaporated milk), 23.3% vs. 48.6%; formula, 25.3% vs. 18.6%; fruit juice, 12.3% vs. 8.6%; sweetened water/tea, 8.2% vs. 7.1%, overall, *p* = 0.001. High, but decreased, addition of sugar or sweetener to bottles, 34.5% in high-intensity vs. 55.4% low-intensity group, *p* < 0.001. NS difference in current or ongoing two-handled cup use. Earlier initiation of two-handled cup: 13.9 ± 0.6 months vs. 15.7 ± 0.7 months, *p* = 0.048.
Maguire et al. 2010. Canada.	To determine education for parents of 9 month infants can reduce bottle use and anaemia at 2 years of age.	RCT; 2006–2007.	Healthy 9 month infants, attending routine paediatrician health visit. IG: *n* = 129, 126 at 15 mo, 102 followed up at 2 years. CG: *n* = 122, 99 followed up at 2 years.	IG: Education, during 9 month visit: iron-deficiency anaemia risk, tooth decay, limit milk talk. Repeat education at 15 month if child still using cup. Resources distributed: sippy cup, bottle cessation protocol. CG: Standardised nutrition counselling: iron-rich first foods, solid food feeding, food safety, limit fruit juice.	At 2 years: Decreased bottle use during day: 15% in IG vs. 40% in CG, *p* = 0.0004. Decreased bottle use in bed: 3% in IG vs. 10% in CG, *p* = 0.05. Earlier cup use: median 9 months in IG vs. 12 months in CG, *p* = 0.001. Earlier bottle cessation: median 12 months in IG vs. 16 months in CG, *p* = 0.004. NS difference in daily milk and juice intake.
Jamieson et al. 2018. Jamieson et al. 2019. Smithers et al. 2017. Australia.	To investigate whether a culturally appropriate multi-faceted oral health promotion intervention reduced Aboriginal children’s intake of sugars from discretionary foods at 2 years of age.	Parallel RCT; February 2011–May 2012.	South Australian mothers, pregnant or giving birth to a baby with Aboriginal Australian ethnicity within 6 weeks. IG: *n* = 223 mothers, 224 infants—159 received intervention; 148 children followed up at 2 years. *n* = 225 mothers, 230 infants—165 received intervention; 145 children followed up at 2 years.	IG: Motivational interviewing and education at pregnancy, 6 months, 12 months and 18 months infant age, with home visits: oral health, diet, dental treatment, fluoride varnish, caries aetiology from sugary foods and drinks, reduce sugary and cariogenic foods, no SSBs in bottles at night. Goal planning for addressing barriers. CG: Delayed intervention, at 24 months, 30 months and 36 months child age.	At 2 years: Increased infant weight z-score: 0.9 in IG vs. 0.6 in CG, *p* = 0.019. Decreased infant height z-score: −0.2 in IG vs. −0.5 in CG, *p* = 0.028. NS difference in BMI z-score; prevalence of overweight or obesity; % energy intake from sugar in discretionary foods; % energy intake from discretionary foods. Decreased ECC prevalence: 19.7% in IG vs. 23.6% in CG, *p* < 0.0001. At 5 years: NS difference in ECC prevalence.
Schroth et al. 2015. Canada.	To determine the effectiveness of the Healthy Smile Happy Child community project on promoting early childhood oral health. To assess changes in carer knowledge, attitudes, and behaviours relating to early childhood oral health, and the burden of ECC and severe ECC.	Serial cross-sectional evaluation of a community program; 1999 start.	4 First Nation communities in Manitoba, Canada: one remote, one rural, two urban. IG: *n* = 319 children evaluated. CG: *n* = 408 children evaluated pre-program.	IG: Community capacity building. Embed oral health promotion in community activity, child and family programs and services. Health promotion education: child comforting without bottle, water intake, bottle cessation and cup transition, avoid cariogenic drinks, no bottle propping, no sweeteners on pacifiers; with development of teaching tools and resources. CG: N/A.	After 5 years: Increased disagreement to ‘frequently giving my child fruit juice or drink is OK for their teeth’: 70.9% at follow-up vs. 48.3% pre-program, *p* < 0.0001. Decreased agreement to ‘frequently giving my child milk or formula is OK for their teeth’: 67.1% at follow-up vs. 74.3% pre-program, *p* = 0.015. NS change in disagreement to ‘frequently giving my child soda is OK for their teeth’: 97.2% at follow-up vs. 94.3% pre-program. Increased disagreement to ‘as my baby gets older, they should use a bottle whenever they want’: 72.1% at follow-up vs. 61.8% pre-program, *p* = 0.0022. Increased disagreement to ‘it is OK to put my baby to bed with a bottle’: 79% at follow-up vs. 70.3% pre-program, *p* = 0.0073. Increased agreement to ‘bottle feeding after my child is one year old is bad for their teeth’: 78.1% at follow-up vs. 62% pre-program, *p* < 0.01. Increased disagreement to ‘babies who do not have bottles will cry more’: 64% at follow-up vs. 54.3% pre-program, *p* = 0.014. NS difference in bottle feeding prevalence. Increased sippy cup use: 93% at follow-up vs. 77.8% pre-program, *p* = 0.0001. NS difference in ECC prevalence. Significant difference in ECC prevalence across 4 community sites, *p* = 0.0012: increased ECC in rural and remote communities, c.f. to one urban community. Trend towards decreased severe ECC prevalence: 38.6% at follow-up vs. 45% pre-program, *p* = 0.08. Significant difference in severe ECC prevalence across 4 community sites, *p* = 0.0052: increased ECC in rural and remote communities, c.f. to one urban community.
Sgan-Cohen et al. 2001. Israel.	To measure the effect of a community health education program on reported infants’ bottle-feeding practices and toothbrushing behaviour, with or without distribution of toothpaste and toothbrushes.	Quasi-experimental 2 × 2 comparison study; dates NR.	Mother and child health centres, providing services to 6–12 months infants. Stratified by religion profile: secular-moderately religious mixed or predominantly Orthodox. IG 1: health education & resource distribution, *n* = 268. IG 2: health education, *n* = 187. CG 1: usual care & resource distribution, *n* = 133. CG 2: usual care, *n* = 139.	IG: Education at each visit, between 0–2 years: decrease SSB frequency, cup transition, tooth cleaning, avoid added sugar, avoid bottle use as pacifier or sleeping with bottle, dental attendance, dental pamphlet. Resource distribution: toothpaste, toothbrush, at baseline, 2 months and 4 months visit. CG: Usual care, with or without resource distribution.	Use of bottle feeding decreased from age 6–12 months to 12–18 months; however, adding sugar to bottles increased. NS difference in use of bottles with added sugar during meals at 6 months, 10.2% in IG vs. 6.3% in CG, *p* = 0.06. However, this increased from 1.5% in IG vs. 1.3% in CG at baseline, *p* < 0.001. Less use of bottle with added sugar between meals at 6 months, 42.4% in IG vs. 47.3% in CG, *p* = 0.06. However, this increased from 20.6% in IG vs. 20.9% in CG at baseline, *p* < 0.001. NS difference in use of bottles during meals; bottles between meals; bottles during sleep; bottles with added sugar during sleep.
Strippel et al. 2010. Germany.	To examine the effectiveness of expanding and improving oral health education in a clinical setting.	Prospective controlled trial; July–Dec 2001.	Parents of ~6 week (IG 1) or ~24 month (IG 2) children, attending routine paediatric examination. IG 1: *n* = 1015. IG 2: *n* = 1025. CG: age-matched children in northern Germany. CG 1: *n* = 1181. CG 2: *n* = 989.	IG 1: Education at 6 weeks and 7 months: breastfeeding, bottle feeding, tooth eruption, oral hygiene, fluoride supplements, nursing bottle use. IG 2: Education at 24 months: caries prevention, oral hygiene, nursing bottle use, drinks in bottles, fluoride supplement. Education to cover 7–8 oral prevention topics, with 15 min education. CG: Conventional oral health education.	For IG 1 (aged 7 months): NS difference in bottle use in bed at night. Decreased bottle use at daytime: ‘never’, 33% in IG vs. 24% in CG; daily, 32% in IG vs. 41% in CG, *p* < 0.001. Decreased ‘sometimes or always’ adding sugar to pureed baby food: 24% in IG vs. 32% in CG, *p* < 0.001. Increased oral prevention topics addressed by clinicians: 3.3 ± 2.1 topics vs. 1.9 ± 1.7, *p* < 0.001. For IG 2 (aged 24 months): NS difference in bottle use at daytime and in bed; ongoing nursing bottle use; frequency in cariogenic food at daytime and in bed; frequency in cariogenic beverage use in bed; knowledge of cariogenic foods and juice. Decreased frequent cariogenic beverage use at daytime: 61% in IG vs. 66% in CG, *p* = 0.013. Increased oral prevention topics addressed by clinicians: 4.2 ± 2.2 topics vs. 2.4 ± 1.7, *p* < 0.001.
Ventura et al. 2021. USA.	To assess effect of policy, systems and environmental change strategies to promote responsive bottle feeding on RWG risk.	Cluster-RCT. May-August 2019 recruitment; 2020 follow-up interrupted by COVID-19 pandemic.	Mothers with new-born infants in USA WIC program. IG: 124 mother–infant dyads. CG: 122 mother–infant dyads.	IG: Policy, systems and environmental change of WIC program. Retooling of infant feeding assessment to be inclusive of responsive bottle feeding; development of assessment tools, counselling probes resources; responsive bottle-feeding online education and text message support; rebranding of infant feeding helpline. CG: Usual care, including timely and tailored breastfeeding support; breastfeeding support resources (helpline, online education, responsive text message support).	At 6 months: Decreased RWG: OR: 0.36 (0.16–0.81), *p* = 0.014. NS difference in exclusive bottle feeding: 65% in IG vs. 65% in CG. NS difference in mixed breast milk and formula feeding: 16% in IG vs. 13% in CG. NS difference in responsive feeding style; pressuring feeding style; encouragement of bottle emptying; percentage of daily feeding from a bottle.
Vichayanrat et al. 2012. Thailand.	To report effect of a multi-level oral health intervention pilot on carer’s oral health practices.	Pilot quasi-experimental trial; dates NR.	Carers of healthy 6–36 months children in 4 sub-districts across 2 provincial districts. IG: 62 carer-child dyads. CG: 52 carer-child dyads.	IG: Community mobilisation. Oral health home visits every 3 months. Improved oral health education, including tooth brushing, bottle feeding, controlling cariogenic intake, and services delivered during child vaccination. CG: Routine health services and toothbrushes from local health centres.	NS difference in prevalence of caries; children falling asleep with bottle; sweetened milk, juice or soda in bottles; consumption of all snacks, in IG and CG. Increased carer knowledge on not putting juice in bottles: 66.1% in IG vs. 34.6% in CG, *p* = 0.001.
Weber-Gasparo, Reeve, et al. 2013. Weber-Gasparo, Warren, et al. 2013. USA.	To compare whether a videotaped message, informed by the self-determination theory, leads to greater changes in oral health knowledge and behavioural intentions to prevent childhood caries.	RCT; dates NR.	Mothers with child aged 12–49 months, in USA WIC program. IG: *n* = 283; 155 completed 1 month follow-up QNR; 181 completed the 6 month follow-up visit. CG: *n* = 132; 78 completed 1 month follow-up QNR; 86 completed the 6 month follow-up visit.	IG: Education: 15 min video on oral health, informed by self-determination theory re: caries aetiology, oral hygiene, diet and caries risk, early caries identification. CG: Neutral-language paper brochure, with same dental content.	At 6 months follow-up: NS difference in ECC prevalence; sippy cup use at night with non-water drinks; sippy cup use at daytime with sugary drinks; sugary drinks consumed between meals; daily intake of >6 oz 100% fruit juice; daily intake of >2 cariogenic snacks; intake of 100% juice; intake of drinks with added sugar; intake of all sugary drinks (added sugar drinks and 100% fruit juice), in IG and CG. Increased maternal oral health knowledge (including bottle use in bed, cariogenic foods and drinks) from baseline: 5.17 ± 3.90 in IG vs. 3.11 ± 4.25 in CG, *p* < 0.001.

AA: African American, BMI: body-mass index, CG: control/comparator group, ECC: early childhood caries, FF: formula feeding, HCP: health care professional, HV: health visitor, IG: intervention group, NHS: United Kingdom National Health Service, NR: not reported, NS: not statistically significant, N/A: not applicable, OR: odds ratio, oz: ounce/s, QNR: questionnaire, RCT: randomised controlled trial, RWG: rapid weight gain, SDS: standard deviation score, SSB: sugar sweetened beverages, WIC: Women, Infant’s and Children’s USA federal assistance program and WFL: weight-for-length.

## Data Availability

Data sharing is not applicable to this article, as no new data were created or analysed in this study.

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
