# Peer review of "Interventions Targeting Bottle and Formula Feeding in the Prevention and Treatment of Early Childhood Caries, Overweight and Obesity: An Integrative Review"

_ijerph, 2021, doi:10.3390/ijerph182312304_

Round 1

Reviewer 1 Report

This is an integrative literature review of studies that addressed best practices in formula feeding and bottle feeding to modify dental caries to determine if there were concurrent effects on overweight and obesity and vice versa.

The paper provides appropriate background to explain the authors' reasoning for combining these outcomes. The literature search process was appropriate and well described. 

The authors described the results and limitations of the described studies appropriately and gave reasonable goals for future studies. 

This is an interesting and well written paper that will add to the literature and likely inspire additional research.

Author Response

We thank reviewer #1 for their feedback, with no further to action.

Reviewer 2 Report

The present paper is an interestingly approaching an actual issue; it exhaustively tries to cover all aspects regarding the “obesogenic and cariogenic feeding behaviours, such as increased sugar exposure through bottle propping and overfeeding”under the form of a narrative (non-systematic) review (“all-inclusive overview’ as author mentions) screening about 27 research papers, with the main conclusion that “Many studies were at risk of bias in study design. All studies focused on carer education; .. only 10 studies utilised behaviour change techniques or theories that addressed antecedents to obesogenic or cariogenic behaviour”

Title: Interventions targeting bottle and formula feeding in the prevention and treatment of early childhood caries, overweight or obesity: an integrative review

The present study objective was (a) “to identify interventions, trials or programs undertaken to support best-practice formula feeding or bottle cessation in infants and children, and (b) to examine their effectiveness on formula feeding practice, bottle cessation, dental health and/or child weight outcomes”.

The authors have done a good job as they have reviewed most of the existing literature in the field on a general topic of high interest for clinical practice, research, and academia, but hard to read. However, some issues must be improved as follows.

  1. Keywords

dental caries; pediatric obesity; bottle feeding; infant formula; infant health; cariogenic agents; health education; child health services; preventative health services; diet, cariogenic

Keywords are numerous (over 10) and redundant (cariogenic agents, cariogenic). Please review

  1. Abstract
  • Please review the English language correctness and character number
  • The type of research, the accessed databases (methodology) are insufficient presented here (in brief)
  • Approached topics may be better delineated (as described in the body of the paper)
  • Conclusions might reflect briefly the research purpose mentioned in the title

  1. References
  • Most of the references are actual, however, 40 of 100 papers were older than five years.
  • Please check the correctness of the reference format

  1. Introduction
  • Introduction not mentioning what is not known in the area (knowledge gap). Line 106-108 approaches this aspect - but in the materials methods section - may be better fitted here

  1. Materials and methods
  • Table 1,2 may be better fitted as supplementary, a narrative presentation of the inclusion and exclusion search criteria may be more suitable

Major review

  • Prisma chart must mention the nine databases used (CINAHL, eMBASE, GH, ProQuest, MEDLINE, Pubmed (Pubmed is an interface used to search Medline and other databases, could you comment on this why searched both?), Web of Science (WOS is not a database, but a meta-search engine) Please clarify the explored research databases and please add to the PRISMA flowchart.
  • Prisma flowchart may be up to 50% minimized to enhance lisibility
  • The screened papers period is missing (last 10, 5 or alltime?)
  • In the inclusion/exclusion criteria there is information that the authors have not specified: Were doctoral theses included? Were meta-analyzes, Clinical cases included? What type of studies were searched (all type?)
  • Reverse search was performed?
  • City and country of the databases, programs used must be inserted
  1. Results
  • Table 3 and Figure 2 my be fitted in Supplementary/reduced – to enhance the readability
  • Tables should be more concise or re-designed to better keep in one page the results. It is difficult to get an idea when the Table is expanded into two-three pages.
  • A graphical illustration of the results may improve the quality of the interesting paper and to better serve the readership, but would be the choice of the authors
  •  

  1. Conclusions

The conclusion of the review might answer to the topics in the title: what interventions targeting bottle and formula feeding showed benefit in prevention and treatment of early childhood caries, overweight or obesity – in the sitematically reviewed literature.

  1. Discussions
  • please concise the main result of the study followed by your recommendations derived from the study results
  • it is recommended not to mix the discussions with the conclusion (based on study results)
  •  Discussion section may follow the order of the topics approached in the results section, to be easier to follow.

Author Response

Please see the attachment - we respond to Reviewer 2 in the attached document.

Reviewer 3 Report

The systematic review deals with an important public health issue namely the infant feeding behaviours that impact on both risk of developing dental caries and risk of obesity.

The review has many strengths:

  • The authors make an important point that although the same infant feeding behaviours (e.g. bottle feeding), impact on both dental caries and risk of obesity, most research is conducted in silo’s and there is little integration of the two fields. Hence this is important contribution to the current literature.
  • The review summarises well the available research in the area and emphasizes the lack of good quality studies.
  • The manuscript is balanced and well written.
  • The research questions on page 3 are succinct and well described.
  • Table 3 is excellent and summarises all studies appropriately.
  • A key point in the discussion (paragraph 3 on page 3) that many studies focus on caregiver educational without using behavioural change techniques is important.

There are few weaknesses in the review other than those acknowledged in the study limitations.  However, I think it would help the reader if the author’s recommendations (for current practice and future research) could be succinctly and clearer set out in the concluding paragraph.  

Author Response

We thank reviewer #3 for their feedback.

With regards to the feedback about the Conclusion: We have edited the Conclusion based the feedback of all reviewers. Please see page 32, lines 585 to 602.

Round 2

Reviewer 2 Report

Paper quality has further improved